# Line-Features-Based Pose Estimation Method for the Disc Cutter Holder of Shield Machine

**DOI:** 10.3390/s23031536

**Published:** 2023-01-30

**Authors:** Zhe Xie, Guoli Zhu, Dailin Zhang, Dandan Peng, Jinlong Hu, Yueyu Sun

**Affiliations:** School of Mechanical Science and Engineering, Huazhong University of Science and Technology, Wuhan 430074, China

**Keywords:** visual positioning, rounded edge model, contour extraction, line detection

## Abstract

To achieve automatic disc cutter replacement of shield machines, measuring the accurate pose of the disc cutter holder by machine vision is crucial. However, under polluted and restricted illumination conditions, achieving pose estimation by vision is a great challenge. This paper proposes a line-features-based pose estimation method for the disc cutter holder of the shield machine by using a monocular camera. For the blurring effect of rounded corners on the image edge, a rounded edge model is established to obtain edge points that better match the 3D model of the workpiece. To obtain the edge search box corresponding to each edge, a contour separation method based on an adaptive threshold region growing method is proposed. By preprocesses on the edge points of each edge, the efficiency and the accuracy of RANSAC linear fitting are improved. The experimental result shows that the proposed pose estimation method is highly reliable and can meet the measurement accuracy requirements in practical engineering applications.

## 1. Introduction

A shield machine is a kind of large construction machine commonly used in tunneling construction, such as subways, highways, and railroads [1]. During the shield tunneling process, the shield machine head rotates and drives the disc cutters to cut up the rock. However, the disc cutters are prone to wear and tear during tunneling, and replacing them is an expensive and time-consuming activity that affects the speed and efficiency of shield tunneling [2,3]. The existing disc cutter replacement method is manually done and needs assisted mechanical tools. The working conditions are hazardous, especially in some special construction environments [4]. In the case of a mud-water balanced shield, the safety risk of disc cutter replacement work is high as the construction workers have to enter the high-humidity and high-pressure work bin. Therefore, the automation of disc cutter replacement is very necessary and has become one of the important research directions in the field of shield construction [5,6]. During automatic replacement, visual positioning is usually selected as a high-precision positioning method to locate the disc cutter holder accurately.

In practice, due to the narrow replacement space, the camera has very limited permissible mounting space and must be mounted on the end of the disc cutter replacement robot. Multiple cameras can obtain a lot of information but have insufficient mounting space, and the accuracy of the depth camera that conforms to the mounting dimensions cannot meet the requirement. On the contrary, the monocular vision measurement mode provides a feasible method for the visual system in the disc cutter replacement robot.

The vision positioning algorithm is divided into two steps: key features acquisition and feature-based pose calculation. The key features on the object must be simple and easy to identify, such as points [7,8], circles [9], lines [10,11], etc. The disc cutter holder has typical line features but no obvious corner points, and in reality, there may be a certain amount of mud on the surface. Thus, how to extract linear features quickly and accurately from the image under pollution is an important issue for the research. Existing line detection methods can be divided into two categories: one is to obtain the linear features directly from the gradient information in the grayscale image, such as LSD [12], Linelet [13], etc; the other is to obtain edge points through image preprocessing and then extract the straight lines on the target boundary by combining the methods of straight line recognition, such as Hough transformation line detector [14], etc. The former cannot handle relatively complex edge cases and is particularly sensitive to noise; in contrast, the second method is more effective in processing complex images. To obtain image edge, the commonly used methods are edge detectors [15,16] and image-segmentation methods to extract contours. For edges with uneven illumination and blurred boundaries, the traditional edge detection operators cannot extract edges well. The commonly used sub-pixel edge extraction methods to handle blurred edges rely more on the characteristics of the image edge itself and do not consider some specific factors that cause edge blurring, such as rounded corners. Among the image segmentation methods, the region-growing method [17] is the most commonly used, but it is less efficient in processing high-resolution images.

Since the edge points on each edge of the disc cutter are easily separated, fitting a straight line to each edge becomes a two-dimensional data fitting problem, which is faster than global detection methods such as probability Hough transform [18], EDlines [19], Cannylines [20], etc. The commonly used methods are the least square method and the RANSAC fitting method [21]. The RANSAC algorithm-based linear fitting method has higher robustness and accuracy, but the disadvantage is that the running time is more affected by the number of outlier points and the efficiency is lower.

For features-based pose calculation, there are many mature methods, including Perspective-n-Point (PnP) methods [22,23,24,25] and Perspective-n-Line (PnL) [26,27,28] methods. For the disc cutter holder positioning, Peng proposes a PnP method that aims to minimize the sum of the chamfering distances [29,30], which must intersect the corresponding feature lines to obtain the feature points. However, in practice, each extracted edge may not be complete, resulting in additional errors in the process of finding the intersection points.

The contributions of this paper are shown as follows: first, to address the blurring effect of rounded corners on image edges, we propose a rounded edge model to obtain edge points that better match the 3D model of the workpiece by the edge search box. Second, we propose a contour separation method based on an adaptive threshold region growing method to obtain the edge search box corresponding to each edge under uneven illumination. Third, by performing some preprocessing on the edge points of each edge, we improve the efficiency and accuracy of the RANSAC linear fitting method.

The rest of this paper is organized as follows. Section 2 states the hardware framework of the vision measurement system. Section 3 describes our proposed linear feature extraction algorithm. Section 4 shows the result of the simulation of the state-of-the-art PnP method and PnL method on this problem. In Section 5, experiments are implemented to test the feasibility of the practical engineering application of our method. In Section 6, the conclusions are given.

## 2. Introduction to Hardware Framework

Figure 1 shows the schematic diagram of the disc cutter replacement robot and the installation position of the vision system. For the convenience of the study, we use the hardware framework shown in Figure 2 in the laboratory. In the framework, the pose-estimation module, which consists of a calibrated camera and an LED array light source, is mounted at the end of the robotic arm. Due to the extremely limited mounting space allowed on the tool-changing robot, a smaller LED array light source is selected for lighting. Considering the completeness of the features, the inner contour of the disc cutter holder with line features was selected for positioning. In practice, each edge may have clay and defects, and the effects of these factors also need to be taken into account in the design phase. How to accurately extract the linear features in the image is a key step in pose estimation. The pose information is transmitted to the robot cabinet, which controls the robot manipulator to complete disc cutter replacement. The requirement for position accuracy must be within 1mm, and the required accuracy for attitude must be within 1°.

## 3. Proposed Method

The flow chart of our proposed method is shown in Figure 3. First, we acquire the image of the disc cutter holder from the calibrated camera. After Gaussian filtering and image brightening, we downsample the image and use the region growing method with an adaptive threshold to obtain the inner contour and then separate the contour of each edge. Then, the corresponding edge search box is generated based on the contour of each edge. Among the edge search lines in the edge search box, the edge points that better match the model are obtained after processing the one-dimensional grayscale curve on the search line based on the rounded edge model. For all edge points on an edge, a preprocessing-based RANSAC linear fitting algorithm is used to obtain the linear feature. Finally, we estimate the pose based on the 3D model of the disc cutter holder with linear features of the image by using the MinPnL algorithm. The rounded edge model, the edge search box generation method, and the preprocessing-based line fitting algorithm of our proposed method are detailed as follows.

### 3.1. Rounded Edge Model

To accurately obtain the line features of each edge of the inner contour of the disc cutter holder, it is necessary to extract the edge points. As shown in Figure 4, the traditional edge detector could not extract the edge of the disc cutter holder image well. There are two reasons, one is that the gradient of each part of the image varies greatly due to uneven illumination, which is not convenient for the separation of the edge points of the edges. The other is that the rounded corners on the edges make the gradient of the image maximum within a certain pixel width (about 15 pixels), and the location of the true edge points cannot be obtained directly from the image.

Due to the effect of the light source and rounded corners, the edges on the image of the disc cutter holder can be divided into two types: strongly illuminated edges and weakly illuminated edges, which present different edge characteristics, respectively. As is shown in Figure 5a, the grayscale change at the weakly illuminated edge shows an overall trend of a step-down, close to the ramp model in the edge model, but due to the existence of noise (such as scratches on the workpiece surface), where the fastest gradient change is not the real edge point. The one-dimensional grayscale curve of 400-pixel points near a strongly illuminated edge is shown in Figure 5b. The grayscale curve at the strongly illuminated edge shows an overall trend of abrupt change, close to the roof model in the edge model within a certain width (10 pixels), where the point with the largest grayscale value cannot be considered as the true edge point.

To obtain more accurate edge points from the image, we construct two edge models with rounded corners as follows.

#### 3.1.1. Strongly Illuminated Edge Model

In Figure 6, the plane to be measured is Π. In the actual measurement, the imaging plane of the camera and Π is close to parallel, so each key point can be projected onto the imaging plane to reflect the situation. In this case, the camera picks up the light reflected by the rounded corners, which is reflected in the image as a higher luminance value than the adjacent area. Note that the brightest point on the corner of the circle is $P_{2}$, the projection point of which on Π is $P_{2}{}^{\prime }$. The center of the corner is $O_{r}$, the starting point is $P_{1}$, the end point is $P_{3}$, and the projection point of $P_{3}$ on Π is $P_{3}{}^{\prime }$, and the radius of the corner is r. When there is no rounded corner, the theoretical edge point is noted as $P_{0}$. The key points on Π correspond to the grayscale changes as shown in the figure. Where $P_{1}$ corresponds to the starting point of the grayscale increase, $P_{2}{}^{\prime }$ corresponds to the maximum grayscale value, $P_{3}{}^{\prime }$ corresponds to the endpoint of the grayscale decrease, and $P_{0}$ corresponds to somewhere in the grayscale decrease. $O_{c}$ represents the center of the camera, and Q is the intersection of the optical axis and Π. The distances from $O_{c}$ to Q, and from $P_{0}$ to are b and a, respectively. In the established coordinate system $O_{c}-\mathrm{xy}$ the coordinates of the following points can be obtained as:(1)$$P_{1}\left(a+r,b\right),P_{3}\left(a,b+r\right),P_{3}\left(\frac{\mathrm{ab}}{b+r},b\right),P_{0}\left(a,b\right)$$

With these coordinates, the length relationship is calculated as:(2)$$\frac{\bar{P_{1}P_{0}}}{\bar{P_{0}P_{3}{}^{\prime }}}=\frac{b+r}{a}$$

Denote the coefficient α as the ratio of the distance between the real edge point and the edge start point to the length of the edge segment. Due to b>a>>r,α can be written as:(3)$$A=\frac{\bar{P_{1}P_{0}}}{\bar{P_{1}P_{3}{}^{\prime }}}\approx \frac{b}{a+b}$$

In the actual measurement, a≈200mm, b≈600mm, α≈0.75. Therefore, the point in the one-dimensional grayscale change curve located at the edge of the strong illumination, whose distance from the edge start point is 0.75 times the width of the edge segment, is considered to be the actual edge point.

#### 3.1.2. Weakly Illuminated Edge Model

Similarly, as shown in Figure 7, the plane to be measured at the disc cutter holder is noted as Π. When the reflection of the round corner is away from the camera, the camera will not receive the light from the light source reflected at the round corner, which is reflected in the image as a ramp edge. The center of the corner is $O_{r}$, the starting point is $P_{1}$, the endpoint is $P_{3}$, and its radius is r. The center of the camera is $O_{c}$, the line $O_{c}P_{2}$ is tangent to $\overset{⌒}{P_{1}P_{3}}$ at $P_{2}$, and $P_{2}{}^{\prime }$ is the projection point of $P_{2}$ on Π. When there is no rounded corner, the theoretical edge point is identically noted $P_{0}$. The corresponding grayscale changes of these key points on Π are shown in the figure, where $P_{1}$ corresponds to the starting point of the grayscale increase, $P_{2}{}^{\prime }$ corresponds to the endpoint of the grayscale decrease and $P_{0}$ corresponds somewhere in the grayscale decrease. $O_{c}$ represent the center of the camera, and Q is the intersection of the optical axis and Π. The distance from $O_{c}$ to Q is b, and the distance from $P_{0}$ to Q is a. A coordinate system $O_{c}-\mathrm{xy}$ is established. According to the geometric relationship, $∠O_{c}P_{2}{}^{\prime }Q=\alpha$, $∠P_{1}P_{2}{}^{\prime }O_{r}=∠P_{2}P_{2}{}^{\prime }O_{r}=\theta$, where α+2θ=π. Let $\bar{P_{1}P_{2}{}^{\prime }}=x$, we can reach:(4)$$\frac{b}{a-r+x}=\frac{2\cdot \frac{r}{x}}{{\left(\frac{r}{x}\right)}^{2}-1}$$

Equation (3) can be expanded as:(5)$$\left(b+2r\right)x^{2}+2r\left(a-r\right)x-{\mathrm{br}}^{2}=0$$

This is a quadratic equation and is easy to solve. After eliminating the negative solution, x is calculated as:(6)$$x=\frac{-r\left(a-r\right)+r\sqrt{{\left(a-r\right)}^{2}+b\left(b+2r\right)}}{\left(b+2r\right)}$$

Due to b>a>>r,x can be written as:(7)$$x\approx \frac{-a+\sqrt{a^{2}+b^{2}}}{b}r$$

Denote the coefficient α as the ratio of the distance between the real edge point and the edge endpoint to the length of the edge segment. α is calculated as:(8)$$\alpha =\frac{\bar{P_{2}{}^{\prime }P_{0}}}{\bar{P_{1}P_{2}{}^{\prime }}}=\frac{r-x}{x}=\frac{a+b-\sqrt{a^{2}+b^{2}}}{\sqrt{a^{2}+b^{2}}-a}$$

In the actual measurement, a≈200 mm, b≈600 mm, α≈0.387. Therefore, the point in the one-dimensional grayscale change curve after the end of the grayscale descent section, whose distance from the edge endpoint is 0.387 times the width of the edge segment, is considered to be the actual edge point.

#### 3.1.3. Edge Search Box

To summarize, the edge model with rounded corners consists of two types: one is the roof model when the camera can receive light reflected from rounded corners, and the actual edge point is in the grayscale descent section; the other is the ramp model when the camera cannot receive light reflected from rounded corners, and the actual edge point is at the lower level of grayscale value after the grayscale descent section. The position of the actual edge point is related to the length of the grayscale edge segment, the working distance, and the distance between the edge and the camera’s optical axis.

As shown in Figure 8, to obtain all edge points on an edge side, an edge search box is used to search the edge points by searching lines inside. For each search line, the one-dimensional grayscale variation curve is obtained from the starting position to the ending position, and the edge point on the search line is calculated according to our proposed rounded edge model after median filtering. By setting several parallel search lines in the search box, we can reach all the edge points on the edge side. Meanwhile, there is a certain search interval between two adjacent search lines to improve the search efficiency. The effect of edge point extraction on one of the edges is shown in Figure 9, and the result shows the rounded edge model can accurately extract the edge points on the image of the disc holder.

### 3.2. Edge Search Box Generation Method Based on Improved Region Growing

To obtain the position of the edge search box corresponding to each edge, it is necessary to make a rough estimate of the position of each edge of the disc cutter holder. There is a large grayscale difference between the surface to be measured and the background part. As a common method of image segmentation, the region-growing method is used to separate them. In the case of high image resolution, the region-growing method takes a longer time. Therefore, to improve the processing efficiency, we downsample the original image and reduced the image resolution from 4112 × 3008 to 411 × 300 pixel size (the region growth time was reduced from 300 s to 1 s). The contour is roughly at the center of the image, so we can directly set the image center point as the seed point. After the seed point is selected, the most critical thing is the selection of the growth threshold. In the image of the disc cutter holder, due to the large size of the holder and uneven illumination, the grayscale distribution of the disc cutter holder surface is different, so it is difficult to select a fixed threshold to separate the inner contour of the disc cutter holder, and the adaptive growth threshold method can better meet the image processing requirements.

We adopt an adaptive threshold region growth method based on the local grayscale average, which appropriately increases the threshold level when the local gray average near the point to be grown is large and decreases the threshold level when the local grayscale average is small to ensure the integrity of the extracted disc cutter holder internal contour. The specific steps are shown in Table 1.

As shown in Figure 10, there are a few areas that cannot be processed well by the fixed threshold. On the contrary, the inner contour is extracted completely well by the adaptive threshold based on the local grayscale average. After testing, the best extraction effect is achieved when $T_{base}$ is set to 2.

After extracting the inner contour of the disc cutter holder, we need to separate the contour of each edge. Since the shape of the inner contour of the disc cutter holder is relatively regular, we can first separate the horizontal edge points from the vertical edge points in the direction of the edge points to segment each edge. We determine whether the edge point belongs to the horizontal or vertical edge point based on the sum of the absolute value of the gradient difference between the X and Y directions of the adjacent points of each edge point. After separating the edge points by direction, however, due to the presence of interference, the contour of each edge does not consist of individual vertical or horizontal edge points but also requires the edge connection to form a complete edge. After finding each edge, it is easy to find the smallest rectangle that can contain all the edge points of this edge, and the search box corresponding to each edge is gotten after expanding the rectangle along the shorter edge. The specific steps are shown in Table 2.

Figure 11 shows the result of our proposed edge search box generation method in a binary image of the disc cutter holder’s inner contour. Figure 11b and Figure 11c show the horizontal and vertical contour points for each edge, where every edge is well separated. The search boxes for each edge are shown in Figure 11d. The result indicates that this algorithm can automatically generate edge search boxes very well.

### 3.3. RANSAC Linear Fitting Algorithm Based on Preprocessing

The RANSAC algorithm is a commonly used method for iterative robust estimation of model parameters from a data set, which can better reduce the influence of outliers on the results. However, the iteration number of the RANSAC algorithm is closely related to the approximate proportion of outliers in the data, so it is necessary to preprocess the data to eliminate the more obvious wrong edge points to improve the efficiency of the algorithm.

Unlike the general random number of data points to fit a straight line, in the previous step, we used the edge search box tool to obtain the edge points; the interval of each edge point is fixed and known, so the obtained edge point data can be treated as a linear sequence, and the value of each item is the distance from the starting line. Let the sequence of edge points as $EP=\left[X_{1},X_{2},\ldots ,X_{n}\right]$, and it is processed based on three types of processing: Continuity Processing, Linearity Processing, and Co-linear Processing.

A.Continuity Processing

For the edge points on each edge of the disc cutter seat, the type of contamination is the presence of defects and mud obscuration on the edge of the seat itself. Regardless of the type of contamination, it manifests itself as a continuous interval effect in the edge point sequence. The purpose of the continuous interval treatment is to remove abruptly varying discrete points from the edge points. The processing steps are shown in Table 3.

B.Linearity Processing

Since the shape of the continuous interval formed by the wrong edge points is generally irregular and cannot be well fitted to a straight line, it can be used as a judgment criterion to determine the type of the continuous interval. The processing algorithm is shown in Table 4.

Compared to the exact linear fitting that follows, $LD_{min}$ can be set a little larger here.

C.Co-linear Processing

Since clay shade and edge, defects break the edge sequence into several linear intervals, erroneous linear intervals generally do not have linear intervals that are co-linear with them, only intervals that have co-linear features with others will be saved. The processing algorithm is shown in Table 5.

The processing effects of each step are shown in Figure 13. After continuity processing, most of the mutation noise is removed. Multiple edge segments caused by clay and defect are eliminated by linearity processing. Finally, after co-linear processing, we obtain the correct edge point of the inner contour of the disc cutter holder.

Figure 14 shows the flow chart of the RANSAC linear fitting method based on pre-processing. The iteration time and fitting error results are shown in Table 6, which indicate that our proposed preprocessing method can effectively reduce the iteration time and improve the accuracy of the RANSAC algorithm on this problem.

## 4. Simulation for Pose Calculation Algorithms

After obtaining the linear features of each edge of the tool holder, to calculate the pose of the disc cutter holder, there are two options: one is to use the linear features directly to calculate the pose by the PnL algorithm; the other is to intersect the lines to reach the corresponding feature points and then calculate the pose by the PnP algorithm. To compare the solution accuracy of the PnP algorithm and the PnL algorithm, we simulate the state-of-the-art algorithms ASPnP and MinPnL by adding random Gaussian noise and masking a part of the straight line.

In the simulation experiment, a virtual camera with a resolution of 4112 × 3008 pixels is synthesized with a focal length of 8 mm and a pixel size of 1.85 μm. The motion range of the disc holder in space is [−10°, 10°] in all three rotation angles, [0 mm, 50 mm] in the X direction, [−50 mm, 50 mm] in the Y direction, and [500 mm, 700 mm] in the Z direction. First, we randomly generate a pose in the above pose space and obtain the image coordinates of the 20 straight lines of the internal contour of the tool holder. Considering the actual situation, all lines in the image are randomly shortened (to ensure that the total length of the retained lines is not less than 0.6 times the length of the original complete line), and random noise is added at the end points of the lines under the normal distribution, where the variance of the noise increases from 0 to 5 pixels. After obtaining all intersection points, we calculate the absolute error in six degrees of freedom. The test is repeated 500 times. The final error is expressed as the standard deviation of all results.

The results are shown in Figure 15. In all six degrees of freedom, the error of the ASPnP algorithm is larger than the MinPnL algorithm. It is worth noting that in some cases, the ASPnP algorithm has larger solution errors due to intersection calculation errors, whereas the MinPnL algorithm has higher stability in comparison. Therefore, we choose the MinPnL algorithm for calculating the pose.

## 5. Experimental Results

### 5.1. Line Detection Experiment

During construction, the high-pressure water gun will be used to wash the disc cutter and its holder which will be replaced. However, the disc cutter holder cannot be completely rinsed off, and there might still be clay left on it to obscure the line features. In order to simulate the state of clay obscuring the disc cutter holder, we use soil to randomly block the internal contour of the knife holder in our experiments. This method is easier and can simulate worse real conditions. Figure 16 shows a comparison of the line detection results between the classical CannyLines and EDLines methods and our method. The first column is the original image, the second column is the result of CannyLines, the third column is the processing result of EDLines, and the fourth column is the result of our method. The second and third columns show that the traditional CannyLines and EDLines methods cannot extract the complete straight lines in our problem, and bring a large number of unrelated straight lines, which will greatly affect the accuracy and efficiency of the detection. In contrast, our method extracts all complete straight lines under no pollution and light pollution conditions, and even under heavy pollution, our method still accurately extracts a larger number of complete straight lines. The last three rows show that our method can also achieve good extraction results in various poses.

### 5.2. Accuracy Verification Experiment

To verify the accuracy of the pose measurement method we proposed, we use a laser tracker to calibrate the accuracy. Due to the heavy weight of the disc cutter holder, it is impossible to change its position and attitude. Therefore, we installed the vision measurement system at the end of the Yaskawa robot to change the three-dimensional position and attitude of the camera. The laser tracker we use is Leica AT901-MR (Unterentfelden, Switzerland), the uncertainty of which in the range of 1.5 m is 0.024 mm (0.015 + 0.006 mm/m). The robot we use is Motoman-gp7. Our camera is MER-1220-9GM/C (Beijing, China), the resolution of which is 4024 × 3036, and the pixel size is 1.85 μm × 1.85 μm. The focal length of the lens is 8 mm. Before the experiment, the instinct matrix of the camera has been calibrated by the checkerboard.

As shown in Figure 17, the relationship between the camera coordinate system $O_{C}$ and the laser tracker coordinate system $O_{L}$ can be expressed by a homogeneous transformation matrix Mcamlaser as:(14)Mcamlaser=[RT01]

Then, we have:(15)$$P_{i}^{laser}=R\cdot P_{i}^{cam}+T$$
where $P_{i}^{cam}$ is the three-dimensional coordinate of the spherical prism center in the camera coordinate system, and $P_{i}^{laser}$ is the three-dimensional coordinate of the spherical prism center in the laser tracker coordinate system. Since Mcamlaser has twelve entries, at least four points are required to fully determine the system.

As shown in Figure 18, due to the large and bulky size of the disc cutter holder, it is impossible to perform a six-degree-of-freedom transformation, so a camera-assisted motion of the robot arm is used to achieve the relative positional transformation. The transformation matrix Mobjectcami of the coordinate system of the disc cutter holder to the camera coordinate system can be easily measured by the vision measurement method described in the previous section. Since the positions of the disc cutter holder and the laser tracker are not changed during the experiment, the transformation matrix of the disc cutter holder to the laser tracker coordinate system is the same for different camera positions. Due to the fact that there is no suitable feature on the surface of the disc cutter holder to directly measure the conversion matrix, it is impossible to obtain the absolute pose measurement error, so we have to measure the error of the relative pose change of the disc cutter holder as the measurement result by using the invariance of the conversion matrix Mobjectlaser of the camera in two positions. We can express Mobjectlaser as:(16)Mobjectlaser=Mobjectcam1⋅Mcamlaser1=Mobjectcam2⋅Mcamlaser2

Then, the theoretical transformation matrix Mobjectcam2′ in the second position of the camera can be calculated based on Equation (16) as:(17)Mobjectcam2′=Mobjectcam1⋅Mcamlaser1⋅Mcamlaser2−1

The difference in the corresponding pose related to Mobjectcam2 and Mobjectcam2′ is the relative pose error.

Table 7 summarizes the entire process of the accuracy verification experiment.

Figure 19 shows the self-built setup. The estimation error of the rotation matrix R is shown in Figure 20a. The rotation matrix R can be defined by three angles: α (roll), β (pitch), and γ (yaw), which are the rotation angles around the Z, X, and Y axes, respectively. The α-error is between −0.6° and 0.6°, the β-error is between −0.3° and 0.3°, and the γ-error is between −0.1° and 0.1°. The translation error is shown in Figure 20b, the x-error is between −0.6 mm~0.9 mm, the y-error is between −0.7 mm~0.6 mm, and the z-error is between −0.8 mm~0.7 mm. Overall our measurement method has an attitude measurement accuracy of 0.6° and a displacement measurement accuracy of 0.9 mm, which meets the accuracy requirement.

In general, for the roof edge, the highest gray value is generally taken as the edge point, while for the ramp edge, the midpoint of the two endpoints of the ramp is generally taken as the edge point. For the results collected from the accuracy verification experiments, we conduct the pose calculation by conventional edge processing and by our proposed rounded edge model, respectively. As shown in Table 8, the result indicates that the absolute value of the maximum error of the results with conventional edge processing is greater than that with rounded edge model processing in all six directions of freedom, and the runtime of our method is shorter. Furthermore, the results of conventional edge processing cannot meet our accuracy requirements, which proves that our rounded edge processing is necessary.

## 6. Conclusions

In this paper, we propose a monocular vision method based on line features to estimate the pose of the disc cutter holder. After taking pictures of the disc cutter holder with a CCD camera, the region growing method with adaptive growth threshold is used to extract the contour under uneven illumination, and then through edge point classification and edge connection, the corresponding edge search box is generated. For the blurring of borders caused by rounded corners, the accurate edge points on each edge are obtained by the rounded edge model and the edge search box. After obtaining all edge points of each edge, the RANSAC algorithm with preprocessing is used for line fitting, where the results show that the preprocessing can significantly improve the accuracy and efficiency. For the pose calculation method, the simulation results show that the MinPnL algorithm is more suitable for our problem. Accuracy verification experiments based on the laser tracker with relative positional errors show that the pose estimation method proposed in this paper can meet the required positioning accuracy and our rounded edge model can better extract the edge points that match the physical model.

The results indicate that our proposed method has good stability and reliability, and can solve the problem of visual positioning of disc cutter holders under contaminated conditions, which helps to realize industrial automation. Our method can also be used in other cases of positioning workpieces containing rectangular contours. For accurate extraction of linear features with rounded edges, our proposed rounded edge model combined with edge search boxes will also be helpful. The measurement accuracy can be further improved in the positional calculation algorithm, including the improvement of the vanishing point-based positional calculation method and the improvement of the PnL algorithm. Developing positional measurement methods for multi-rectangular combinations of targets is our future research focus.

## Figures and Tables

**Figure 1 sensors-23-01536-f001:**
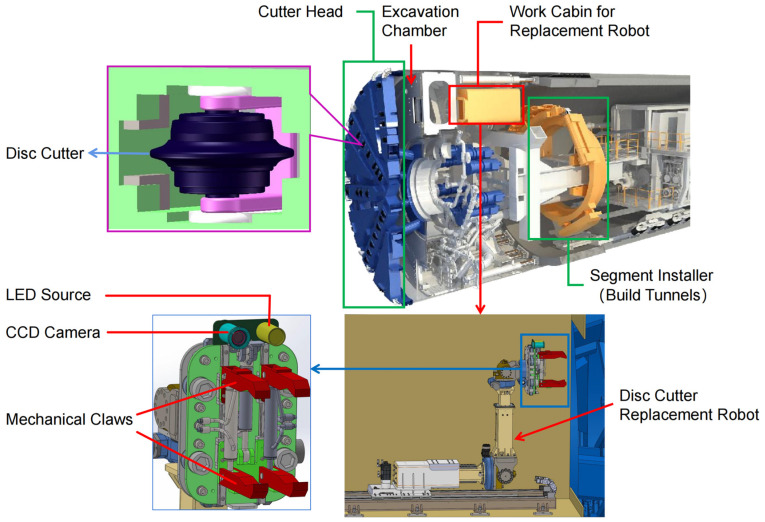
Schematic diagram of the disc cutter replacement robot.

**Figure 2 sensors-23-01536-f002:**
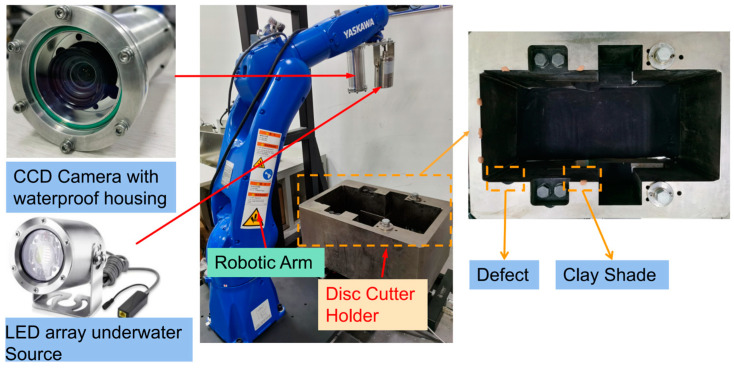
The hardware framework of the pose-estimation module in the lab.

**Figure 3 sensors-23-01536-f003:**
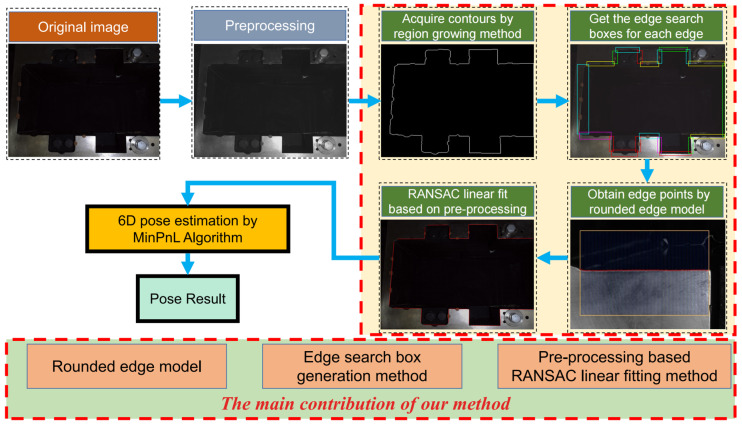
The flowchart of our proposed method.

**Figure 4 sensors-23-01536-f004:**
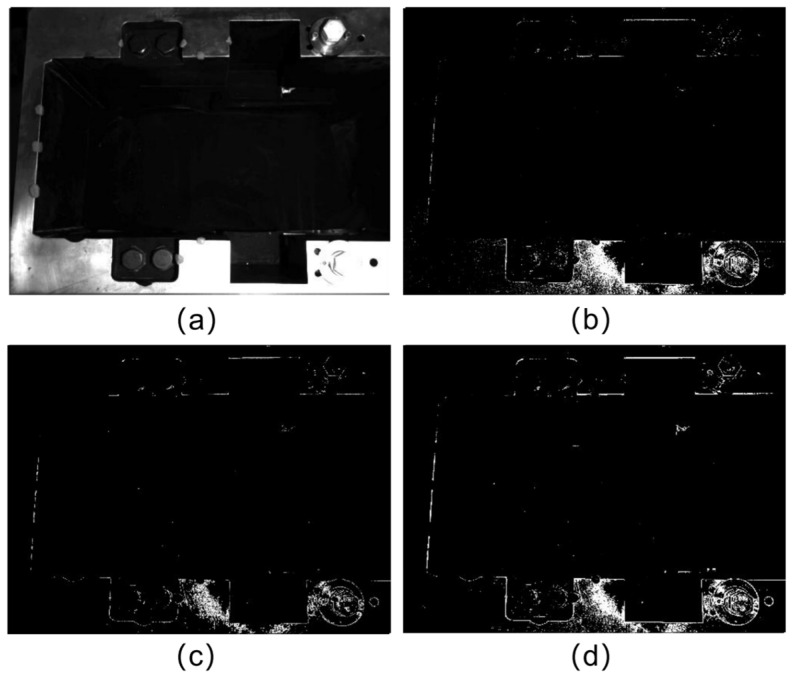
Results of edge detector: (**a**) Original image; (**b**) Laplacian’s Outcome; (**c**) Canny’s Outcome; (**d**) Sobel’s Outcome.

**Figure 5 sensors-23-01536-f005:**
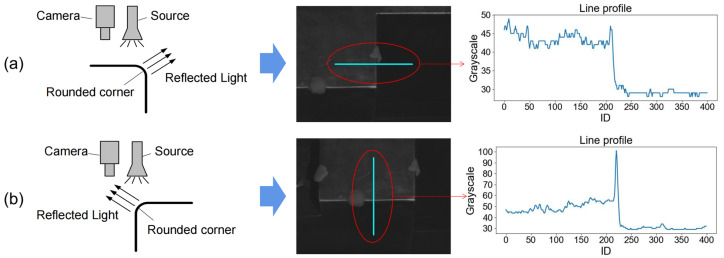
The line profiles of two edge types: (**a**) Weakly illuminated edge; (**b**) Strongly illuminated edge.

**Figure 6 sensors-23-01536-f006:**
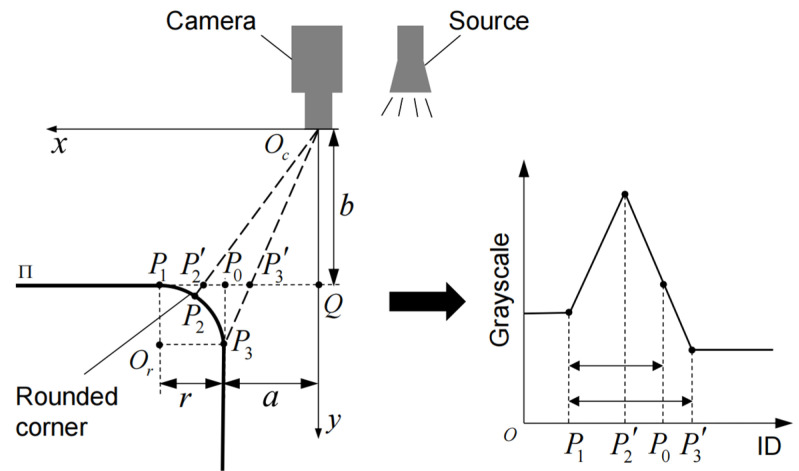
Schematic diagram of strongly illumination edge model.

**Figure 7 sensors-23-01536-f007:**
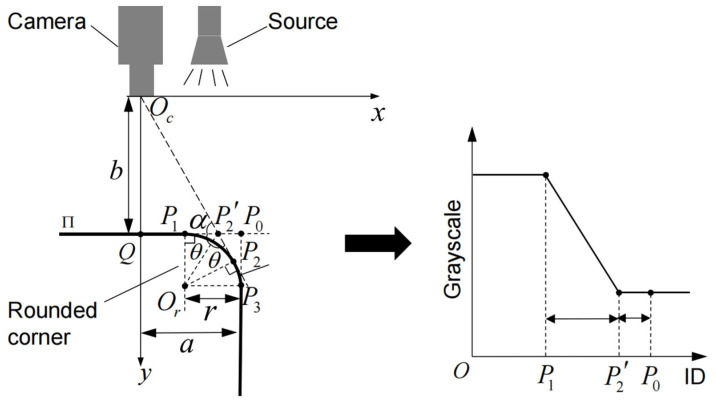
Schematic diagram of weakly illumination edge model.

**Figure 8 sensors-23-01536-f008:**
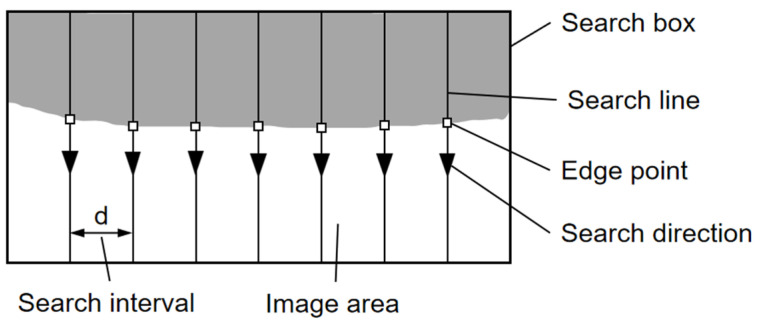
Schematic diagram of the edge search box.

**Figure 9 sensors-23-01536-f009:**
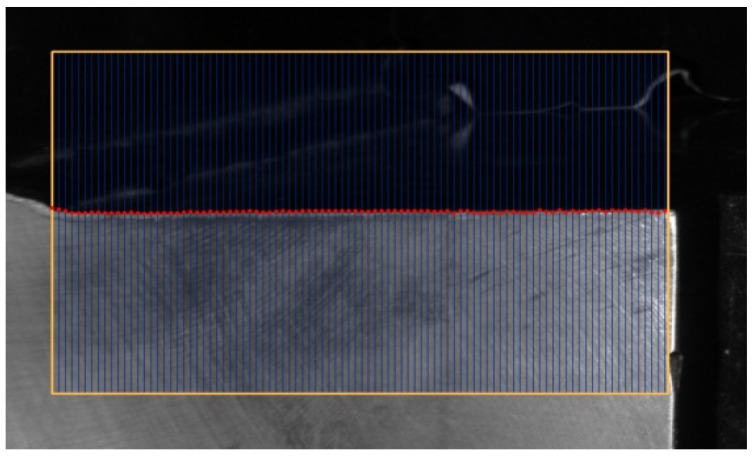
The effect of edge point extraction.

**Figure 10 sensors-23-01536-f010:**
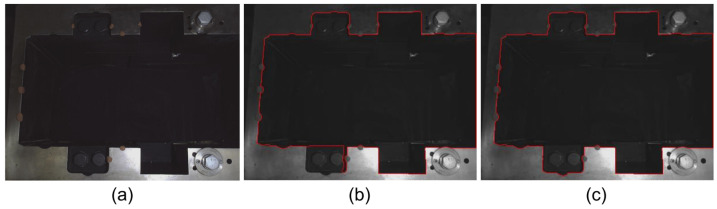
Results of region growing methods (**a**) Original image; (**b**) By fixed threshold; (**c**) By adaptive threshold based on the local grayscale average.

**Figure 11 sensors-23-01536-f011:**
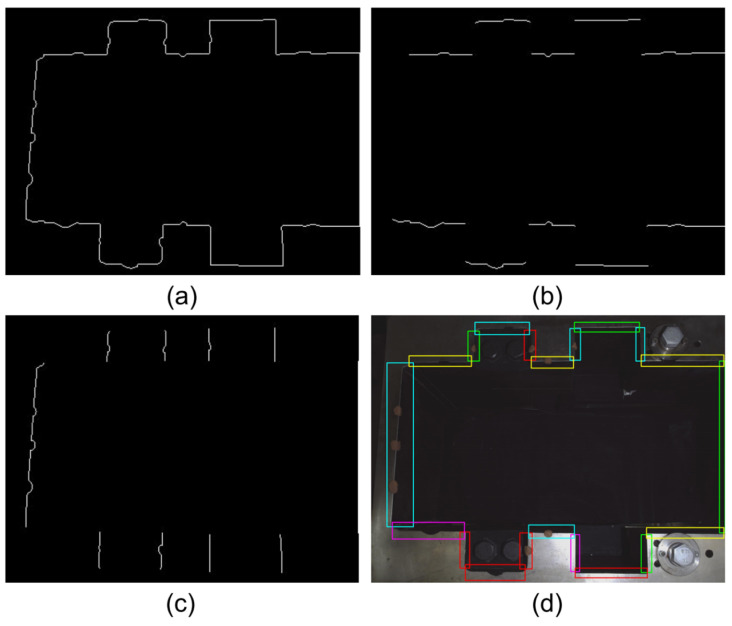
Results of contour separation and edge search box generation: (**a**) Original contour; (**b**) Contour points for horizontal edges; (**c**) Contour points for vertical edges; (**d**) Edge search boxes.

**Figure 12 sensors-23-01536-f012:**
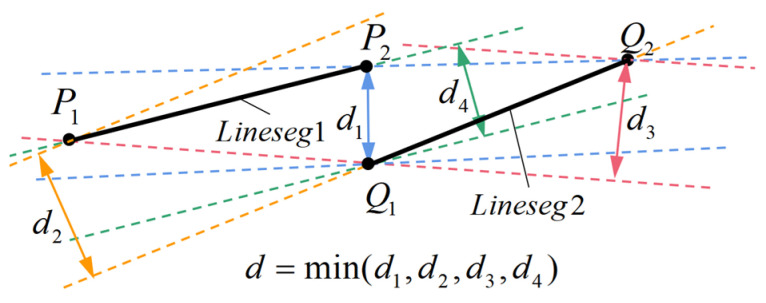
Diagram of Line Segment Distance.

**Figure 13 sensors-23-01536-f013:**
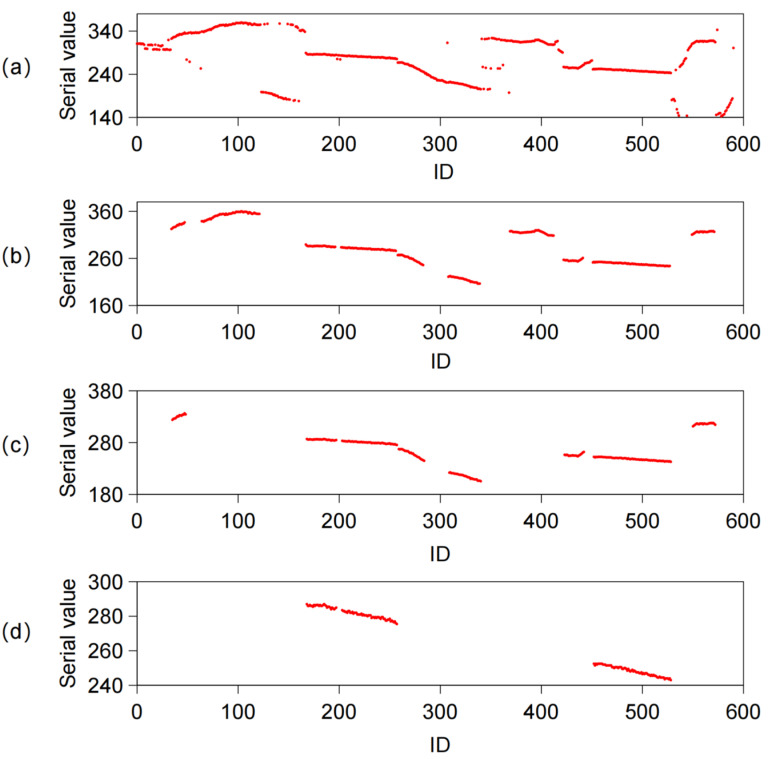
Results of line fitting preprocessing: (**a**) Original sequence; (**b**) Continuity Processing’s Outcome: (**c**) Linearity Processing’s Outcome: (**d**) Co-linear Processing’s Outcome.

**Figure 14 sensors-23-01536-f014:**
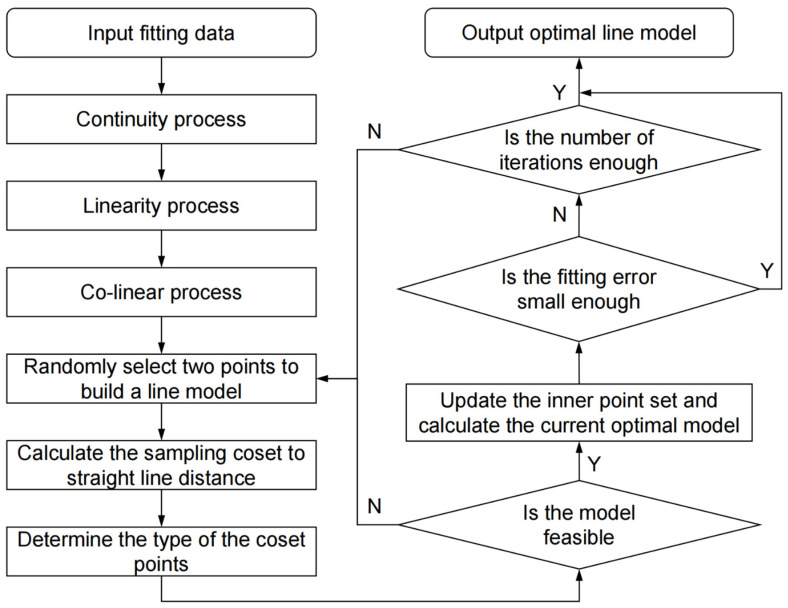
Flow chart of RANSAC linear fitting based on pre-processing.

**Figure 15 sensors-23-01536-f015:**
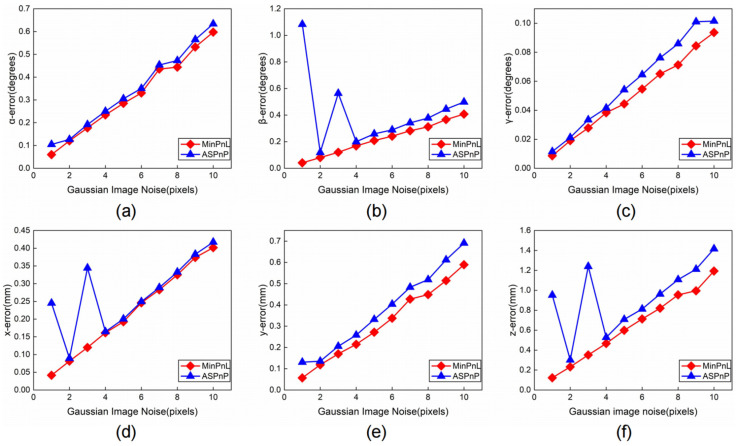
Accuracies of the MinPnL algorithm and the ASPnP algorithm when noise level changes: (**a**) The rotation error in roll (α) direction; (**b**) The rotation error in pitch (β) direction; (**c**) The rotation error in pitch (γ) direction; (**d**) The translation error in X-axis direction; (**e**) The translation error in Y-axis direction; (**f**) The translation error in Z-axis direction.

**Figure 16 sensors-23-01536-f016:**
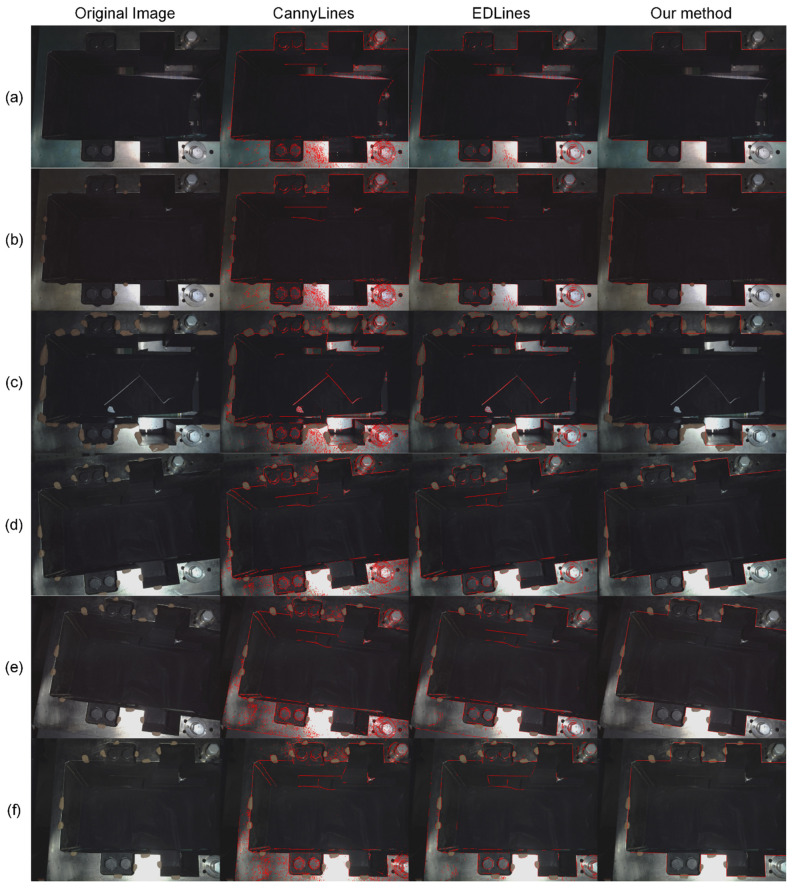
Results of line detection experiment under different conditions: (**a**) No pollution; (**b**) Light pollution; (**c**) Heavy pollution; (**d**–**f**) Different poses.

**Figure 17 sensors-23-01536-f017:**
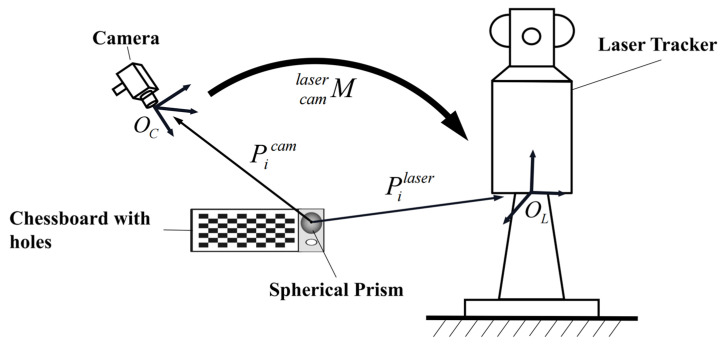
The schematic diagram of coordinate system conversion.

**Figure 18 sensors-23-01536-f018:**
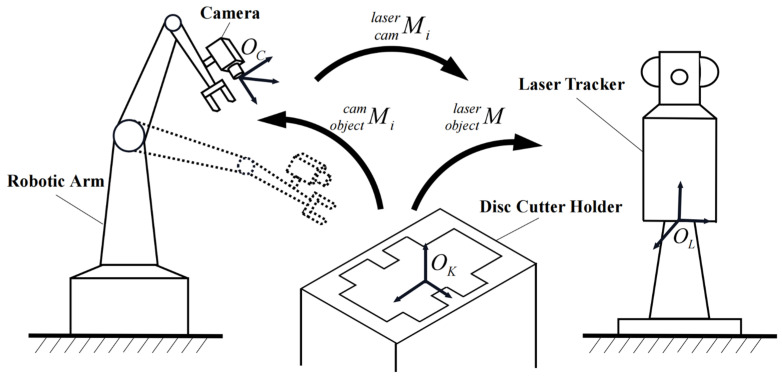
The schematic diagram of relative pose measurement.

**Figure 19 sensors-23-01536-f019:**
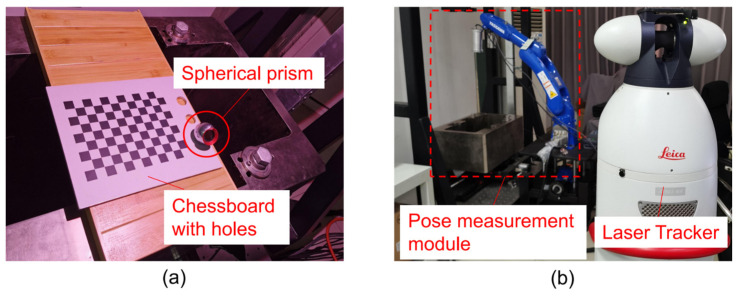
The setup of the accuracy verification experiment: (**a**) Spherical prism and checkerboard; (**b**) Laser tracker and pose measurement module.

**Figure 20 sensors-23-01536-f020:**
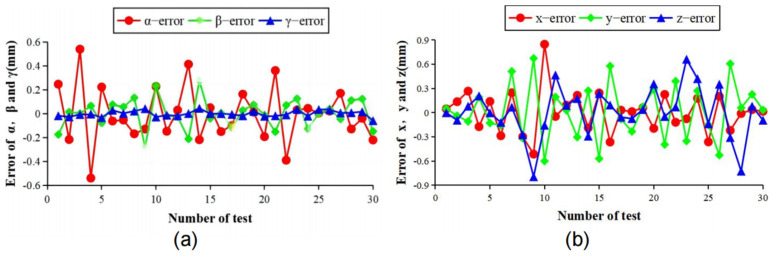
Accuracy verification experiment results: (**a**) Error of α, β, and γ; (**b**) Error of x, y, and z.

**Table 1 sensors-23-01536-t001:** Steps of the contour extraction method based on improved region-growing.

**Input:**	The grayscale image of the disc cutter holder
**Step1:**	Set the image center point to the seed and add it to the sequence Q to be grown.
**Step2:**	Take a point $S_{b}$ from Q and calculate the mean gray value $G_{m}$ of a square pixel area centered on $S_{b}$. The growth threshold T at $S_{b}$ is set as (9) $$\begin{array}{c} T=\left(\frac{G_{m}}{30}+1\right)*T_{base} \end{array}$$ Calculate the absolute value of the difference $G_{diff}$ between the grayscale of each ungrown point in the eight-neighborhood of $S_{b}$. If $G_{diff}\leq T$, add the ungrown point to Q. After calculating all eight-neighborhood points, add $S_{b}$ to the growth completion sequence E.
**Step3:**	Repeat Step (2) if there are remaining points in Q.
**Step4:**	After the growth is completed, mark all points in E as 1 and other points as 0 in a new binary image to reach a background binary image.
**Step5:**	Extracts the outer contour of the background binary image, which is the inner contour edge of the disc cutter holder
**Output:**	The inner contour edge of the disc cutter holder

**Table 2 sensors-23-01536-t002:** The steps of the rectangular search box generation method.

**Input:**	The inner contour edge of the disc cutter holder
**Step1:**	For each edge point $P_{i}\left(X_{i},Y_{i}\right)$ in the inner contour, calculate the sum of the absolute values of the gradient difference in the X direction as: (10) $$\begin{array}{c} D_{x}=\sum_{i=-2}^{1}\left|X_{i+1}-X_{i}\right| \end{array}$$ and also calculate the sum of the absolute values of the gradient difference in the Y direction as (11) $$\begin{array}{c} D_{y}=\overset{1}{\underset{i=-2}{\sum }}\left|Y_{i+1}-Y_{i}\right| \end{array}$$ If $D_{x}>D_{y}$, the edge point is judged as a vertical edge point; otherwise, it is judged as a horizontal edge point.
**Step2:**	Set a distance threshold dis and a count threshold N. If the number of edge points between two edge segments in the same direction is less than dis, as the same edge segment is considered, all edge segments are retained containing more edge points than N.
**Step3:**	Take the leftmost vertical edge section as the starting mark, and mark all the edge sections as the corresponding edge of the disc cutter holder in order.
**Step4:**	For each edge of the disc cutter holder, calculate the rectangle that can completely wrap it, and then expand the shorter side of the rectangle three times to generate the edge search box.
**Output:**	The edge search box of each edge.

**Table 3 sensors-23-01536-t003:** Steps of the Continuity Processing.

**Input:**	The sequence of edge points $EP=\left[X_{1},X_{2},\ldots ,X_{n}\right]$.
**Step1:**	Calculate the sequence value difference between each edge point and the previous edge point as follows: (12) $$\begin{array}{c} X_{m}{}^{\prime }=X_{m+1}-X_{m},m=1,2,\ldots ,n-1 \end{array}$$ Obtain the difference sequence $EP_{D}=\left[X_{1}{}^{\prime },X_{2}{}^{\prime },\ldots ,X_{n-1}{}^{\prime }\right]$.
**Step2:**	For each point in $EP_{D}$, judge whether its absolute value exceeds the given difference threshold $d_{max}$. If $\left|X_{i}{}^{\prime }\right|>d_{max}$, determine it as a mutation point $CP_{i}$, and the edge points between two mutation points $CP_{i}$ and $CP_{i+1}$ form a continuous interval of edge points $CQ_{i}=\left[X_{p},X_{p+1}\ldots ,X_{q}\right]$. The set of all continuous intervals is denoted as $CQ_{ALL}=\left\{CQ_{1},CQ_{2},\ldots ,CQ_{n}\right\}$.
**Step3:**	For each continuous interval $CQ_{i}$ in $CQ_{ALL}$, calculate the interval length of $CQ_{i}$ as $len\left(CQ_{i}\right)=q-p+1$, and determine whether the number of edge points $len\left(CQ_{i}\right)$ included in $CQ_{i}$ exceeds the given length threshold $L_{min}$. If $len\left(CQ_{i}\right)<L_{min}$, remove $CQ_{i}$ from $CQ_{ALL}$. Finally, we can obtain a set of continuous intervals $CQ_{ALL}$ that satisfy the length requirement.
**Output:**	The set of continuous intervals $CQ_{ALL}$.

**Table 4 sensors-23-01536-t004:** Steps of the Linearity Processing.

**Input:**	The set of continuous intervals $CQ_{ALL}$.
**Step1:**	For each $CQ_{i}$ in $CQ_{ALL}$, perform linear fitting $\hat{y}=ax+b$ by least squares with the sequence number of each edge point as x and the sequence value as y.
**Step2:**	Calculate the mean square error (MSE) of the residual of the fitted straight line from each point in $CQ_{i}$ as (13) $$\begin{array}{c} MSE=\frac{1}{n}\sum_{i=1}^{n}{\left(y_{i}-\hat{y}\right)}^{2} \end{array}$$
**Step3:**	Judge whether MSE exceeds the threshold $LD_{min}$, and if $MSE>LD_{min}$, remove $CQ_{i}$ from $CQ_{ALL}$. The lastly retained $CQ_{ALL}$ is denoted as the set of linear intervals $LQ_{ALL}$.
**Output:**	The set of continuous intervals $LQ_{ALL}$.

**Table 5 sensors-23-01536-t005:** Steps of the Co-linear Processing.

**Input:**	The set of continuous intervals $LQ_{ALL}$.
**Step1:**	Calculate the “Line Segment Distance” for every two linear intervals as shown in Figure 12. We define “Line Segment Distance” as the minimum distance between two parallel lines that completely contain two line segments. In practice, the linear interval is approximated as the line segment represented by the two endpoints. For example, note that $P_{1}$ and $P_{2}$ are the endpoints of a line segment $LS_{1}$, and note that $Q_{1}$ and $Q_{2}$ are the endpoints of another line segment $LS_{2}$. If $P_{1},P_{2},Q_{1},Q_{2}$ form a quadrilateral $P_{1}P_{2}Q_{2}Q_{1}$, $d_{11}$ is the distance from $P_{1}$ to the line $P_{2}Q_{2}$, $d_{12}$ is the distance from $Q_{1}$ to the line $P_{2}Q_{2}$, the minimum distance $d_{1}$ of the line $P_{2}Q_{2}$ and its parallel line which can completely envelop two line segments can be written as $d_{1}=\max(d_{11},d_{12})$. Similarly, we can obtain the minimum distance $d_{2},d_{3},d_{4}$ between lines $Q_{1}Q_{2},P_{1}Q_{1},P_{1}P_{2}$ and there corresponding parallel lines that can completely envelop the line segment $LS_{1}$ and $LS_{2}$. Thus, we can obtain the line segment distance $LS_{Dis}$ between $LS_{1}$ and $LS_{2}$ as $LS\_Dis=\min(d_{1},d_{2},d_{3},d_{4})$.
**Step2:**	If the line segment distance $LS_{Dis}$ between two linear intervals is less than the threshold value $CL_{t}$, the two linear intervals are considered to have a common line relationship and are retained, and the linear interval that does not have a co-linear relationship is deleted from $LQ_{ALL}$. Retain the longest interval if the line segment of all linear interval pairs is greater than the threshold value. Save all edge points corresponding to linear intervals in $LQ_{ALL}$ as a good set of edge points $GEP_{ALL}$.
**Output:**	The good edge point set $GEP_{ALL}$.

**Table 6 sensors-23-01536-t006:** The iteration time and fitting error results.

Method	Iteration Time (ms)	Fitting Error (pixel)
RANSAC line fitting	38.251	0.482
RANSAC line fitting with pre-processing	8.810	0.408

**Table 7 sensors-23-01536-t007:** Steps of the accuracy verification experiment.

**Step1:**	Complete the calibration of the intrinsic matrix of the camera before the experiment.
**Step2:**	Randomly move the robot equipped with the vision measurement system to a certain pose within the pose range to be measured.
**Step3:**	Place the target ball on the hole of the calibration plate at six different positions within the field of view. Measure the coordinates of the target ball center in the laser tracker coordinate system and the coordinates of the calibration plate in the camera coordinate system, respectively.
**Step4:**	Calculate the conversion matrix Mcamlaser1 between the camera coordinate system and the laser tracker coordinate system in the measurement system. Calculate the transformation matrix Mobjectcam1 between the disc cutter holder coordinate system and the camera coordinate system.
**Step5:**	Repeat the step1—step4 to obtain the coordinate system transformation matrix Mcamlaser2 and Mobjectcam2 under the second pose of the robot.
**Step6:**	Calculate the theoretical transformation matrix Mobjectcam2′ between the disc cutter holder coordinate system and the camera coordinate system under the second pose of the robot, and calculate the difference between the poses pose2 and $pose2^{\prime }$ corresponding to Mobjectcam2 and Mobjectcam2′ as the measurement error.

**Table 8 sensors-23-01536-t008:** Maximum of the absolute errors of each direction of freedom and runtime.

Method	α-Error (°)	β-Error (°)	γ-Error (°)	x-Error (mm)	y-Error (mm)	z-Error (mm)	Runtime (s)
Our proposed method	0.336	0.291	0.032	0.711	0.704	0.882	1.309
Conventional processing	0.925	0.848	0.230	1.735	1.328	0.976	1.858

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
