# Peer review of "Line-Features-Based Pose Estimation Method for the Disc Cutter Holder of Shield Machine"

_sensors, 2023, doi:10.3390/s23031536_

Round 1
Reviewer 1 Report
The paper presents a study to measure the pose of a disk cutter holder by machine vision.
The paper concentrates on the processing of the monocular image of the tool holder. In my opinion the work done should be better contextualized. How the extracted line features are used to estimate the pose is not well described. Furthermore, I would add some pictures or figures to depict a shield machine, where the tool holder is located, as well as the camera and the tool replacement robot are located. This would make clearer the need of the proposed approach.
I would also expand the state of the art. Are there alternative solutions for the presented problems in other works? Do you see the possibility to apply different approaches, such as 3D scanning using laser sensors or cameras?
The proposed method is well described. My biggest concern is related to the validation phase. It seems the approach is tested on a limited set of images. Furthermore, the test case is built as a laboratory set, with disturbs added artificially. Please, provide data on the performed test cases and an analysis on the robustness of the approach to the varying conditions (real dust/mud, real lighting conditions, accessibility of the scanning volume, …).
How is the approach scalable to other fields or applications? What are the contributions of this work to image vision approaches?
Please improve the quality of the images, they are dark, and it is difficult to see the details.
Author Response
Response to Reviewer 1 Comments:
Point 1: How the extracted line features are used to estimate the pose is not well described.
Response 1: We appreciate it very much for this good suggestion. However, in our method, the pose is calculated using line features based on the well-established PnL algorithm, which calculate the relative pose between the camera and the object from a set of n 3D straight lines and their 2D projected lines with known pixel coordinates. This is described in lines 70-75 of the paper. Since we do not make any improvements to the existing PnL algorithm, but directly use the existing state-of-the-art PnL algorithm to accomplish the estimation of the pose of disc cutter holder after extracting the line features, we do not have a particularly detailed description in this area.
Point 2: Furthermore, I would add some pictures or figures to depict a shield machine, where the tool holder is located, as well as the camera and the tool replacement robot are located. This would make clearer the need of the proposed approach.
Response 2: Thank you for your suggestion. As shown in Figure 1, we have added a picture to the paper including the disc cutter holder, camera and tool change robot location.
Point 3: I would also expand the state of the art. Are there alternative solutions for the presented problems in other works? Do you see the possibility to apply different approaches, such as 3D scanning using laser sensors or cameras?
Response 3: Thank you for your suggestion. Out of the installation size, we do not choose multiple cameras to solve the problem. And in our preliminary experiment, the accuracy of the depth camera that conforms to the mounting dimensions cannot meet the requirement. Thus, we choose the monocular vision. As your suggestion, we add our reasons in lines 40-42.
Point 4: My biggest concern is related to the validation phase. It seems the approach is tested on a limited set of images. Furthermore, the test case is built as a laboratory set, with disturbs added artificially. Please, provide data on the performed test cases and an analysis on the robustness of the approach to the varying conditions (real dust/mud, real lighting conditions, accessibility of the scanning volume, …).
Response 4: We are so grateful for your kind concern. As your suggestion, we have supplemented the analysis of the results of the line detection experiment and also add the results of the line detection at different poses of the camera in section 5.1. For the real light conditions, since there is no other additional ambient light source, we have to rely on the LED array light source for illumination in the actual field. However, due to its uneven illumination, some parts of the picture may be a bit dark, and we also ensure the same lighting conditions as the field during the experiment. In addition, a high-pressure water gun will be used on site to wash down the disc cutter holder, only a small amount of clay may be left on it. Our experiments simulate worse situations than the field environment.
Point 5: How is the approach scalable to other fields or applications? What are the contributions of this work to image vision approaches?
Response 5: Thank you very much for your suggestion. We have added the contribution of our work to image vision approaches in the final conclusion section. Our method can also be used in other cases of positioning workpieces containing rectangular contours. For accurate extraction of line features with rounded edges, our proposed rounded edge model combined with edge search boxes will also be helpful.
Point 6: Please improve the quality of the images, they are dark, and it is difficult to see the details.
Response 6: We gratefully appreciate for your valuable suggestion. As your suggestion, we have improved the quality of the images.
Reviewer 2 Report
The article "Method for estimating the position of the holder of the disk cutter of the panel machine based on linear features" is devoted to improving the accuracy of megerment the position of the holder of the disk cutter using machine vision when replacing the disk cutters of the Shield machines.
The authors note the main problems, including the problem of detecting the edges of the cutting tool, and propose methods to reduce their influence.
The article proposes a linear-based monocular vision method for estimating the position of a disc cutter holder, which shows some improvement in the positioning accuracy of the cutter.
If this obtained accuracy is sufficient, then it is good. But the authors nowhere refer to documents or publications about what accuracy is required. You have to take the word of the authors. It doesn't look very convincing.
As a note to the article, the following should be noted:
1. For unknown reasons, the first formula (164) is not numbered. Consequently, the rest will also require renumbering.
2. In table 1 and in table 4, the formulas are numbered, and in table 3 they are not numbered. Why?
3. In table 6, the dimension of the given parameters is indicated, and in table 8 it is not indicated. I want to understand why the authors do this? This impairs the perception of results.
Author Response
Response to Reviewer 2 Comments:
Point 1: If this obtained accuracy is sufficient, then it is good. But the authors nowhere refer to documents or publications about what accuracy is required. You have to take the word of the authors. It doesn't look very convincing.
Response 1: Thank you for your suggestion. Because our research is based on the National Key Research and Development Plan, this accuracy requirement is proposed by the mission statement, not by ourselves. We're sorry we can't add it to our references.
Point 2: For unknown reasons, the first formula (164) is not numbered. Consequently, the rest will also require renumbering.
Response 2: We feel sorry for the inconvenience brought to the reviewer. As your suggestion, we have renumbered all the formulas.
Point 3: In table 1 and in table 4, the formulas are numbered, and in table 3 they are not numbered. Why?
Response 3: Thank you so much for your careful check. We are very sorry for our negligence about the numbering of formulas in the tables. We have added numbers to the formulas in Tables 2 and 3.
Point 4: In table 6, the dimension of the given parameters is indicated, and in table 8 it is not indicated. I want to understand why the authors do this? This impairs the perception of results.
Response 4: We gratefully appreciate for your valuable suggestion. We have added the runtime parameter to Table 8 as you suggested to ensure consistent dimensionality. Thank you again for your careful check.
Round 2
Reviewer 1 Report
The authors have met the request of the reviewer, even if I would have expect much additional work. However, the requested point have been clarified and the paper is now more clear.
Author Response
It is an honor to have your approval of our work.Thanks for the positive comments.